# Major Depression: One Brain, One Disease, One Set of Intertwined Processes

**DOI:** 10.3390/cells10061283

**Published:** 2021-05-21

**Authors:** Elena V. Filatova, Maria I. Shadrina, Petr A. Slominsky

**Affiliations:** Institute of Molecular Genetics of National Research Centre ”Kurchatov Institute”, 123182 Moscow, Russia; shadrina@img.msk.ru (M.I.S.); slomin@img.msk.ru (P.A.S.)

**Keywords:** major depressive disorder, theories of depression, common mechanisms, etiology, pathogenesis

## Abstract

Major depressive disorder (MDD) is a heterogeneous disease affecting one out of five individuals and is the leading cause of disability worldwide. Presently, MDD is considered a multifactorial disease with various causes such as genetic susceptibility, stress, and other pathological processes. Multiple studies allowed the formulation of several theories attempting to describe the development of MDD. However, none of these hypotheses are comprehensive because none of them can explain all cases, mechanisms, and symptoms of MDD. Nevertheless, all of these theories share some common pathways, which lead us to believe that these hypotheses depict several pieces of the same big puzzle. Therefore, in this review, we provide a brief description of these theories and their strengths and weaknesses in an attempt to highlight the common mechanisms and relationships of all major theories of depression and combine them together to present the current overall picture. The analysis of all hypotheses suggests that there is interdependence between all the brain structures and various substances involved in the pathogenesis of MDD, which could be not entirely universal, but can affect all of the brain regions, to one degree or another, depending on the triggering factor, which, in turn, could explain the different subtypes of MDD.

## 1. Introduction

Major depressive disorder (MDD) is a heterogeneous disease that affects one out of five individuals in their lifetime and is the leading cause of disability worldwide [1]. The symptoms of MDD are associated with structural and neurochemical deficits in the corticolimbic brain regions [2,3,4]. The behavioral symptoms of depression are extensive, covering emotional, motivational, cognitive, and physiological domains [4], and include anhedonia, aberrant reward-associated perception, and memory alterations.

Presently, MDD is considered a multifactorial disease with various causes and triggers such as genetic susceptibility, stress, and other pathological processes such as inflammation. For example, in some cases, genetic factors can promote or even trigger the occurrence of depression [5,6,7,8,9]. Some mutations and polymorphisms can affect the response of receptors to neurotransmitters or biologically active substances [5,10,11,12,13], which, in turn, could affect the resistance of the brain’s chemical balance to stressors. However, it is not yet fully elucidated as to which genes or regions of nuclear or mitochondrial DNA or which types of genetic changes, alone or in combination, can represent reliable genetic markers of depression [14]. Furthermore, the lack of consistent and reproducible findings in genome-wide association studies for MDD can at least partly be explained by the fact that relevant genetic variants confer an increased risk only in the presence of exposure to stressors and other adverse environmental circumstances, i.e., the so-called gene-environment interaction [15,16,17]. In addition, genetic effects are not likely stronger than environmental stressors [15]. Nevertheless, exposure to traumatic or repeated psychosocial and environmental stressors clearly can increase vulnerability to MDD or even cause depressive symptoms in humans [18,19,20,21]. MDD can be spontaneous but often follows a traumatic emotional experience or can be a symptom of other diseases, most often neurological (e.g., stroke, multiple sclerosis, or Parkinson disease) or endocrine (e.g., Cushing’s disease and hypothyroidism) [22]. MDD can also be triggered or precipitated by pharmacological agents or drug abuse [23]. These factors may influence both the overall risk of illness and sensitivity of individuals to environmental adversities. However, in general, the precise causes and mechanisms involved in the etiopathogenesis of MDD are not fully understood.

Numerous studies have been devoted to investigating the causes of depression from the point of view of psychology and psychiatry. Several models of depression have been proposed [24,25,26,27,28,29,30], making a tremendous impact on the psychotherapy of MDD. Most of them were brilliantly reviewed elsewhere [31] and a unified model of depression has been proposed in an attempt to combine the “clinical, cognitive, biological, and evolutionary” aspects of the disease [32].

However, to date, the greatest contribution to the understanding of the pathogenetic mechanisms of MDD has been made by physiological, biochemical, and pharmacological studies. These studies allowed the formulation of several theories that attempt to describe the development of MDD on biochemical, cellular, anatomical, and physiological levels. Such theories include the monoamine hypothesis [33,34,35,36], the stress-induced depression hypothesis [37], the cytokine hypothesis [38,39,40,41,42,43], the neuroinflammation and neuroplasticity hypothesis [18,44,45,46,47,48,49,50,51], the GABA-glutamate-mediated depression hypothesis [44,52,53,54,55,56,57], the circadian hypothesis of depression [6,58,59,60,61], and the cholinergic-monoaminergic interaction theory [62,63,64]. Each hypothesis has its strengths and weaknesses, but they cannot consider and fully describe all the processes and symptoms of MDD (Table 1). Nevertheless, these theories share some common pathways, which lead us to believe that these hypotheses depict several pieces of the same big puzzle. Therefore, in this review, we provide a brief description of all these theories and their strengths and weaknesses in an attempt to highlight the common mechanisms and relationships of all major theories of depression and combine them together to present the current overall picture of etiopathogenesis of MDD.

## 2. The Monoamine Hypothesis

According to the monoamine hypothesis, depressive symptoms [69] occur as a result of altered levels of monoamine neurotransmitters 5-hydroxytryptamine (5-HT)/serotonin [33,70], noradrenaline (NA) [33,71,72], and/or dopamine (DA) [8,73,74,75] (Figure 1). This hypothesis was developed on the basis of multiple evidence that antidepressant therapies increase the neurotransmission tone of one or more of these neurotransmitters [65]. However, several studies demonstrated that the abrupt decrease in the synthesis of 5-HT, DA, or both did not lead to depression in healthy individuals. These findings indicated that concentrations of serotonin higher than a certain threshold are requisite for selective serotonin reuptake inhibitors (SSRIs) in order to be effective antidepressants, leading to the belief that a pronounced depletion of monoamines is not sufficient to cause depression in healthy adults [65,76].

However, this hypothesis does not explain the causes and all of the symptoms of depression, the delayed response to drug therapy, and why antidepressants can only achieve remission, but not complete recovery.

Moreover, there is no clear evidence for one transmitter being central to the etiology of depression. Numerous data suggest that monoamine neurotransmitters are not the only biologically active substances involved in MDD etiopathogenesis [22,77].

## 3. The Hypothesis of Stress-Induced Depression

The hypothesis of stress-induced depression was the first hypothesis that aimed to explain the possible causes of MDD, which were not clarified by the monoamine hypothesis. This theory postulates that the disorder could be causes by chronic stress, and the subsequent malfunctioning of the hypothalamic–pituitary–adrenal (HPA) axis, which is one of the most studied pathological pathways of the pathogenesis of depression [78] (Figure 2). However, the impact of stress depends on the type of the stressing factor, its duration, its genetic background, and its history of life [79]. It is believed that a prolonged and moderate impact of stress could be more dangerous, especially multiple everyday unpredictable disturbing incidents, compared with a single strong stressful impact. It is impossible to adapt to these mild stressful incidents, which continuously stimulate the defense and adaptation mechanisms, leading to their subsequent exhaustion.

The stimulation of the HPA axis, represented by hypophysis, hypothalamus, and adrenal gland in Figure 2, is a key player in those mechanisms [80] and causes the secretion of glucocorticoid hormones, whose function is to provide adaptation to stressors in both the brain and periphery [66]. Glucocorticoids predominantly lead to the redistribution of energy resources and the restoration or defense of homeostasis after a challenge [81].

The elevated activity of the HPA axis in many cases of depression pointed to the probable underlying mechanisms of pathogenesis [77,82,83]. The chronic activation of the HPA axis with continuous stress leads to prolonged alterations in all affected organs and systems [66,84,85,86], which results in the adrenal hypertrophy and thymic atrophy associated with long exposure to corticotropin and elevated glucocorticoid hormone in rats [87]. The combined or single action of excess cortisol and proinflammatory agents could be toxic glial and neuronal cells [65,88,89] and may suppress neurogenesis and neuroplasticity in prefrontal cortex (PFC) and the hippocampus [16,77,90,91], which, in turn, may result in decreased levels of Glu [92,93] and gamma-aminobutyric acid (GABA) [65], cognitive decline [94,95], reduced appetite [8,96], anhedonia [2,97,98], altered cardiovascular tone [99,100], and other symptoms arising from chemical changes in these structures [33,77,84,101,102].

There is a comprehensive neurobiological model that places the HPA axis at the center of development of prolonged consequences of early trauma [16,103] and that hyperactivity of this axis may originate from early life programming [104]. The HPA axis can be sensitized in utero by smoking, maternal stress, early grave loss, and child abuse, all of which could result in a development of MDD later in life [103,105]. However, not every case of early life stress will develop into the disease after new trauma or stress, and not all adults with depression have had early life stress [106]. Severe depression with the overactive HPA axis in some patients is characterized by the hypersecretion of cortisol, the enlargement of pituitary and adrenal glands, and the increased levels of corticotropin-releasing factor (CRF) in the cerebrospinal fluid (CSF), which represent deficits in negative feedback systems and/or excessive central stimulation of the secretion of CRF and/or other substances that promote ACTH secretion [82,101,107]. Overall, numerous studies, reviewed by Willkinson and Goodyer, suggest that a continuous dysregulation of the HPA axis with a central deficit of the feedback mechanisms is predominant in depressive disorders [108]. For example, the increased activity of the HPA system in humans has been associated with glucocorticoid receptor (GR) resistance, which could be the result of either a decreased expression or a reduced functionality of GR [109]. Therefore, ineffective cortisol-mediated negative feedback does not reduce the excessive activity of the HPA axis during chronic stress. Nevertheless, treatment with antidepressants leads to the normalization of the levels of cortisol and CRF via an increase in the expression of GRs in brain, which restore the normal function of the feedback loop [65]. In addition, some data demonstrated the dependence of neurogenic activity of antidepressants from the functioning of GR in human hippocampal cells [110].

Moreover, it was previously shown that 5-HT neurons, which densely innervate the amygdala [111,112], also regulate the HPA [113]. Furthermore, some data point to the possibility that 5-HT decreases the activity of amygdala and may reduce the learning of aversive stimuli [33]. 5-HT may participate in the regulation and control of impulsive behavior [114].

Nevertheless, not all patients with MDD demonstrate an increased function of the HPA axis (hypercortisolism) or a violation of negative feedback in the axis. Thus, the pathological changes of the HPA axis, such as hyper- and hypo-cortisolism, can be used to subtype the disease [103]. However, it remains unclear how hypocortisolism is formed. Nevertheless, it was suggested that prior trauma, which occurs early in life, may be associated with the increased inhibition of cortisol secretion [115]. It seems possible that early-life trauma and continuous stress could elevate the susceptibility of individuals to stress, which may lead to a shift of their cortisol response to greater suppression, and, in turn, make them unadaptable to stress factors [103]. Thus, it is possible that multiple forms of depression with different biochemical profiles exist. Such different subtypes of depression with different abnormalities of the HPA axis may demonstrate the best responses to different treatments [108].

## 4. The Neurotrophic Hypothesis

The neurotrophic hypothesis of depression postulates that a cause and pathogenesis of depression can be explained by a violation of functioning of the neurotrophic system of the brain and the fact that antidepressant treatments may partly result in a reversal of deficiency of this system and lessening of depressive symptoms. The main focus of research on this hypothesis is directed on brain-derived neurotrophic factor (BDNF) [67], which is involved in neurogenesis, regulates differentiation and growth of neurons [116,117,118], as well as other regulators of neuroplasticity, which might affect behavior through their control of neurogenesis. It was suggested that neurogenesis in adults can enhance glucocorticoid-mediated negative feedback on the HPA axis and facilitate resilience to stress [119]; therefore, decreased neurogenesis can be the basis for the development of depression-like symptoms in stressful situations. Indeed, a decrease in BDNF and BDNF pro-peptide levels and expression of BDNF, BDNF-regulated genes, and tropomyosin receptor kinase B (TrkB) have been detected in patients with MDD [120,121,122]. Decreased levels of BDNF and BDNF receptor were also demonstrated in the hippocampus of patients with depression postmortem [117,123,124]. Moreover, the ratio between BDNF-TrkB and proBDNF-p75NTR could also be altered in MDD [22,125]. Furthermore, increased level of cortisol inhibited BDNF, leading to neurodegeneration and partly to the development of symptoms of depression [117,126,127].

In addition, several studies showed that numerous chronic stressors, including administration of corticosterone, which induce depressive and anxiety-like behaviors through the changes in peripheral levels of cortisol and inflammatory mechanisms, can reduce neurogenesis in hippocampus and neuroplasticity in adult animals [16,67,117,128], probably because the prolonged elevation of levels of cortisol is neurotoxic [95]. Similar histological and functional neuroimaging studies demonstrated violations in synaptic and structural plasticity in various brain regions, including hippocampus and the frontal cortex, in patients with MDD [22,129,130]. Chronic stress leads to reduced spine density in the hippocampus, medial PFC, and medial amygdala and increased spine density in the basolateral amygdala in animal studies [131,132]. It was suggested that these changes could interfere with appropriate responses in the brain to adapt to environmental stimuli [133]. On the whole, distinct initial violations of a complex signaling network, such as dysfunction of the HPA axis, deficit of neurotrophic factors, altered expression of microRNAs, abnormal regulation of proinflammatory cytokines, and violated delivery of gastrointestinal signaling peptides may cause a deficit of neurogenesis and result in a similar phenotype, manifesting as major mood alteration. Moreover, all of these factors are interconnected on a functional level, and a primary violation of one of them leads to changes in the others [22,114,128,134]. Therefore, neurogenesis in hippocampus probably plays a part in the normalization of the HPA tone and the regulation of the adequate response of HPA, most likely via negative feedback associated with GR [135].

The neurotrophic hypothesis originated from observations that antidepressants might lessen symptoms of depression by stimulating neurogenesis in adult hippocampus [136] and increase the numbers of adult-born neurons [137,138], forming synaptic connections in mice approximately in 4 weeks [117,139]; this in general, correlates with a lot of evidence showing that classical SSRI therapy achieves efficacy in 3–4 weeks. It was shown that serotonin may positively regulate adult granule cell proliferation and neurogenesis [33,47,112,140,141,142], and both serotonin and BDNF signaling systems participate in the regulation of neural circuitries, the action of antidepressant, and each other [143,144]. Despite multiple evidence demonstrating stress-induced decrease in neurogenesis, several studies, reviewed by Hanson et al., have also shown the lack of correlation between stress and neurogenesis [145]. Therefore, such contradictions suggest that the influence of stress and antidepressants on neurogenesis does not appear in all cases of depression [145].

Overall, the precise role of neurogenesis and BDNF signaling in the pathogenesis of MDD, and whether distinct antidepressant medicines directly affect BDNF and/or serotonin, has not been fully clarified yet [47,119]. Neurogenesis alone cannot explain the etiopathogenesis of MDD, but it may play a part in the development of behavioral and cognitive abnormalities characteristic of depression [146]. Moreover, the concept of a stress-induced inhibition of neurogenesis is probably too reductionistic in entirely explaining such a complex disorder as depression [147], and, most likely, the inhibition of neurogenesis contributes to effects of stress in combination with other mechanisms [67]. Hence, as attractive as the feedforward concept may appear, the concept of a stress-induced suppression of neurogenesis is, as mentioned, most likely too reductionistic in order to completely explain a disorder as complex as depression [147].

In general, it can be concluded that these theories are highly intertwined and based on the same mechanisms that occur in the same organs and tissues (Figure 1 and Figure 2, Table 1). It should be noted that we deliberately simplified the representation of the reward system in order to decrease the complexity of the figures. The functioning of the reward system in MDD is described in detail elsewhere [148,149,150,151].

## 5. The Inflammation/Cytokine Hypothesis

In addition to the previously described processes, other mechanisms of the MDD etiopathogenesis were proposed. They include inflammation [39] and microglial cells [1,152,153,154,155]. It was suggested that the immune system plays an important role in the pathogenesis of the disorder, thus formulating the inflammation/cytokine hypothesis, a simplified scheme, presented in Figure 3. Several clinical and animal studies reported that proinflammatory cytokines could be involved in the development of MDD [39,40,156,157,158,159,160]. Increased levels of proinflammatory biomarkers, such as tumor necrosis factor-alpha, interleukin (IL)-1, IL-6, and C-reactive protein, were detected in the plasma of patients with depressive symptoms, compared with healthy individuals [161,162,163].

A connection of MDD with other pathologies, such as arthritis, asthma, coronary artery disease, diabetes, and obesity, may indicate that inflammation by itself or in conjunction with stress may cause MDD [39]. Depression associated with inflammation is characterized by greater persistence and severity and decreased motivation, and it develops later in life [164,165,166].

The connection between cytokines and depression is supported, to various degrees of strength, by the following generalized conclusions: (1) cytokines administered to patients and laboratory animals induce some symptoms of depression; (2) An activated macrophage/monocyte response of the immune system and elevated cytokine levels were detected in some patients with depression; (3) depressive disorders frequently occur in patients suffering from disorders with an inflammatory component; (4) some stressors induce increased expression of cytokines in both the periphery and the central nervous system (CNS); and (5) some antidepressants have anti-inflammatory properties and can adjust the behavioral responses on an hyperactive immune system [40,156].

Stress may affect cytokines on a genetic level in individuals with a predisposition to MDD [167]. Various stressors, physical as well as psychological, can activate immune system throughout the organism and stimulate the release of inflammatory cytokines [168] that lead to changes of levels of neurotransmitters and behavior [39,40,156,157,158,169,170,171,172,173,174]. Chronic stress or prolonged exposure to inflammatory cytokines results in the glucocorticoid resistance, which can lead to an increased predisposition for the release of other cytokines, such as IL-1beta [175,176,177,178,179,180,181]. The mechanisms of this interplay between the CNS and the organs and tissues have been detected (predominantly in animals) [40,175,182]. Although several studies have shown an interconnection between depression and proinflammatory cytokines, no evidence of high sensitivity or specificity of cytokines to MDD has been found [156]. However, it was reported that antidepressant medicines decrease the release of proinflammatory cytokines from activated immune cells, inhibit chemotaxis, and intensify the synthesis of anti-inflammatory cytokines in humans [183].

Inflammatory cytokines such as interferon-alpha can influence systems and processes that may play an important role in depressogenesis, including the functioning of the frontal lobe and the anterior cingulate [184,185]; the HPA axis [66,186,187,188,189,190,191,192]; the activity of dopaminergic [193,194,195], serotonergic [157,187,196,197,198,199,200,201,202,203,204], glutamatergic [173,197,200,205,206,207], GABA [173], and noradrenergic systems [208,209,210]; the proliferation of hippocampal neurons [211], neurotoxicity [200], neuronal damage and loss of neuronal plasticity [197,211,212,213,214,215,216]; and growth factors [217,218,219]. As a result, the multiple evidence of the imbalance between pro- and anti-inflammatory cytokines leading to the overproduction of neurotoxic metabolites in the brain served as a basis for the proposal of the neurodegeneration hypothesis of depression [203,220].

Importantly, cytokines are large molecules, and circulating cytokines normally do not cross the blood–brain barrier (BBB). However, peripheral cytokines can penetrate into the CNS and activate local immune system by several mechanisms [9,12,162], including (1) passage through leaky regions in the BBB at circumventricular organs [176,221,222] (this passage only occurs with a high concentration of peripheral cytokines [173]); (2) active uptake mechanisms of cytokines across the BBB [223,224,225,226]; (3) local actions at peripheral vagal nerve afferents that transmit signals of cytokines to the appropriate regions of the brain, including hypothalamus (HT) and the nucleus of the solitary tract (the so-called “neural route”) [227,228,229,230]; (4) activation of endothelial cells and perivascular macrophages in the cerebral vasculature to produce local inflammatory mediators such as cytokines, chemokines, prostaglandin E2 (PGE2), and nitric oxide (NO) [231,232,233,234]; and (5) activated peripheral immune cells, which can be recruited to the brain parenchyma and, in turn, produce cytokines in the CNS [235,236]. Signals from peripheral cytokine are amplified in the brain by local inflammatory processes, including pathways of transduction of inflammatory signals, production of cytokines, and release of PGE2 (see Figure 1 for inflammatory pathways in the brain in Felger and Lotrich [12]). Endothelial cells and perivascular macrophages of the brain respond to circulating cytokines by the release of PGE2 and induction of the expression of cyclooxygenase-2 [237,238,239]. Cytokines in the CNS are produced predominantly by microglia but can also be synthesized by astrocytes [240,241], neurons [242,243], and oligodendrocytes [244,245]. Chronic immune activation can transform microglia to synthesize inflammatory mediators that may affect the systems of brain neurotransmitters and the integrity of neurons [12,246]. Activated microglia can produce indoleamine-2,3-dioxygenase and kynurenine-3-monooxygenase, which catabolizes kynurenine [247], inducible nitric oxide synthase [248,249], reactive oxygen and nitrogen species [250,251], and monocyte chemotactic protein-1/chemokine (C–C motif) ligand 2 [252], which is involved in attracting immune cells from periphery into the CNS of mice [235].

In addition, several primary studies and comprehensive reviews (see Table 3 in Liu et al. [39]) made the assumption that dysregulated oxidative and nitrosative pathways [253], as well as mitochondrial dysfunction [254] contribute to depression. Many clinical studies have assessed biomarkers of these pathways in connection with depression [255,256,257,258]. Some findings collectively suggest the existence of a subtype of patients with MDD accompanied by an elevated inflammatory status that leads to unique variations in both etiopathology and clinical presentation [39].

## 6. The Circadian Hypothesis

Although it has been known, since the 1950s, that daily rhythms are disrupted in patients with MDD [6,259,260], the molecular mechanisms linking mood disorders and abnormalities in sleep/wake cycles are still not well understood [117,261]. Nonetheless, robust evidence corroborates a bidirectional link between sleep disturbances and depression, with insomnia now recognized as a predisposing factor for developing depression [6,261,262]. Moreover, it was shown that depression itself can alter sleep structure in numerous ways [263].

Changes in sleep/wake cycles by itself may initiate manifestations of depression [264,265]. Sleep abnormalities may result in relapse and a decreased response to therapeutic interventions [6]. The co-occurrence of depression and abnormal sleep may represent a physiological reaction to a more definitive violation of circadian rhythms, i.e., the circadian disruption could be an antecedent primary condition causing the development of symptoms of depression [266]. Alternatively, sleep disruption and depressive illness may essentially be independent conditions; nonetheless, they may cause reciprocal effects and probably indicate an interference in the feedback processes usually distinguishing their interplay [6].

The circadian theory of depression proposes that stressful events alter schedules of sleep, which, in turn, changes diurnal molecular rhythms in cells, resulting in the development of mood disorder in vulnerable individuals [68,117]. Considering the fact that the sleep/wake and circadian rhythms are closely intertwined, it is therefore not surprising that sleep deprivation therapy (SDT) quickly lessens the intensity of depressive symptoms [117,267]. Previously, it was demonstrated that sleep deprivation affects brain systems involved in emotion, e.g., amygdala [268]. It was shown that the genes controlling circadian rhythms in the anterior cingulate cortex are dysregulated in depression [261], and the neurons in this region increase their activity during sleep and disengagement from tasks [269,270]. Though not yet demonstrated, it is thought that SDT resets the aberrant circadian clock in patients with depression, resulting in alleviation of the symptoms [261,267]. The accumulating clinical evidence highlights potential changes in the circadian clock gene expression in patients with depression. Though limited in number, the few studies on SDT with regard to depression/anxiety have been promising.

A phase advance in cortisol rhythm, another symptomatic feature of depression, was demonstrated in patients with this disorder, especially among those with a melancholic subtype [6]. Lower blood concentrations of melatonin with pronounced circadian phase advances in melatonin secretion are also often observed in patients with MDD [6].

It was demonstrated that ventral tegmental area (VTA) DA neurons are key players in the modulation of behaviors associated with depression [271,272,273], and it is possible that aberration in the expression of circadian genes in the VTA may participate in such behaviors [117] (see also references in Chaudhury et al.). These recent findings also revealed molecular links between the regulation of mood and the circadian timing system, which could become a potential target for the treatment of mood disorders associated with the disturbance of circadian rhythms. The facts that social interaction, a rewarding phenomenon in social animals, is also controlled by circadian rhythms [274], and that depression-like behavior results in changes of neural processing in the reward system, suggest that alterations in the circadian system could lead to abnormal reward processing in the reward center and subsequent behaviors associated with depression [117].

The disturbance of sleep/wake cycles can also be connected to dysfunction in the HT (Figure 3 in Saltiel and Silvershein [275]). The state of awakening is regulated by the sleep/wake switch in HT and monoamine projections from brainstem to the cortex [276]. GABA, histamine, and 5-HT participate in the regulation of normal sleep/wake cycles [275]. Some 5-HT receptors have been associated with circadian rhythm, sleep, and mood [277]. It was established that brain 5-HT synthesis, release, and catabolism are controlled by a diurnal rhythm, and are closely connected with the suprachiasmatic nucleus [6,261]. Serotonergic neurotransmission affects the phosphorylation of CLOCK proteins, which represent the molecular oscillator, leading to shifts of phases and involvement of suprachiasmatic nucleus activity [261,278].

Disturbances in the functioning of orexinergic-locus coeruleus (LC) (noradrenergic)-amygdala circuit may be another probable mechanism of pathogenesis of depression [117]. Neural processing of fear learning has recently been shown to pass from the lateral HT to the amygdala via the LC in rats [279]. Orexin (hypocretin) fibers from the lateral HT were demonstrated to directly depolarize LC neurons via the rapid corelease of Glu and orexin, resulting in the activation of N-methyl-D-aspartate (NMDA) and orexin-1 receptors, respectively [279]. Furthermore, the activation of orexin neurons in LC leads to elevated noradrenergic signaling via beta-adrenergic receptor in the lateral nucleus of the amygdala, resulting in the enhanced formation of fear memory [279].

Disturbance of sleep may also be another variable associated with inflammation [280,281,282] and subsequent higher risk for depression. Sleep deprivation leads to elevated levels of proinflammatory cytokines in blood, when compared with undisturbed sleep [12,283].

However, whether abnormal circadian rhythms can cause depression or whether depression results in the violation of circadian rhythms is still unclear. Nevertheless, there is substantial evidence, both clinical and observational, that a correlation exists between the two, and most individuals with depressed mood also experience irregular circadian rhythm [6]. Thus, circadian dysregulation may be an important pathogenetic component of MDD. A simplified scheme of the processes involved is presented in Figure 4.

## 7. The excitatory Neurotransmitters

As mentioned above, the functioning of GABA and Glu systems also appears altered in depression [56,77]. A simplified scheme is presented in Figure 5. Some studies, reviewed by Hasler et al., demonstrated abnormally decreased plasma and CSF levels of GABA in patients with MDD [284]. Possibly, because 5-HT action across discrete 5-HT receptor subtypes is thought to modulate GABAergic interneurons that influence Glu circuits involved in cognitive functions [285], the changes in 5-HT levels might result in alterations in the levels of Glu, which is essential for cognitive processing. Therapeutic agents that modulate Glu transmission, e.g., memantine and ketamine [286], have demonstrated antidepressant-like properties [287] to the point that ketamine-based drugs were “approved by the FDA for treating of treatment-resistant MDD” [288].

The increased metabolism in limbic thalamocortical neuronal pathways in depression most likely means increased glutamatergic transmission in these pathways [77]. Increased levels of Glu within discrete anatomical circuits may also elucidate the changes precisely in gray matter in mood disorders [84,289]. Magnetic resonance spectroscopic studies also showed alterations of levels of Glu (measured together with cerebral glutamine as the combined “Glx” peak in the magnetic resonance spectroscopic spectra) and GABA in MDD. These data demonstrate the mixed extra- and intracellular pools of GABA, glutamine, and Glu, but the intracellular pools dominate overwhelmingly in these spectra [77]. It was shown that GABA levels were abnormally reduced in the dorsal anterolateral/dorsomedial PFC and the occipital cortex in patients with MDD [284,290]. The greater part of the GABA pool is in GABAergic neurons; thus, the decreased levels of GABA in the dorsal anterolateral PFC are in accordance with the evidence of decreased number of GABAergic neurons in the BA9 area of brain in MDD [291]. Patients with MDD also demonstrate decreased levels of Glx in the ventromedial and dorsomedial/dorsal anterolateral regions of PFC, where neurophysiological and histopathological abnormal changes are detected in depression [289]. Because the Glx levels demonstrate the glutamine and Glu pools inside the cells, the abnormal decrease in Glx levels would be in accordance with the decrease in glial cells discovered postmortem in the those brain regions in MDD, as glia play a prominent part in Glu–glutamine cycling [77].

The hypermetabolism that manifests as elevated metabolism of glucose and is associated with the reduction of gray matter in certain regions of brain, such as PFC, during depression, could indicate an important role of excitatory amino acid transmission in the neuropathology of mood disorders [77].

Changes in the activity of various signaling processes such as BDNF, NMDA, and mammalian target of rapamycin (mTOR) are possible mechanisms that underlie alterations of synaptic plasticity leading to depression [292]. For example, it was shown that stress-induced synaptic deficits in the PFC was accelerated by a primary elevation of Glu release and decreased Glu uptake resulting in increased Glu excitotoxicity and subsequent neuronal atrophy through dendritic retraction, reduced dendritic arborization, decreased spine density, and reduced synaptic strength [292]. Such a violation of synaptic connectivity can potentially result in the decrease in neurotrophic factors such as BDNF, the overall decrease in NMDA signaling, and the inhibition of mTOR signaling that subsequently leads to the manifestation of depression-like behavior [48,117,292]. Thus, a possible mechanism of turnover of depressive behaviors by ketamine-induced NMDA blockade may initially involve the inhibition of presynaptic NMDA receptors at GABAergic interneurons leading to a decrease in inhibitory tone and subsequent net increase in glutamatergic surge, while the inhibition of excitotoxic, extrasynaptic NMDA receptors on the postsynaptic neurons increases cell survival. Furthermore, the increased net glutamatergic surge leads to the increased postsynaptic alpha-amino-3-hydroxy-5-methyl-4-isoxazolepropionic acid receptors’ activation of neuroplasticity-related signaling pathways involving BDNF and mTOR, resulting in overall synaptogenesis and synaptic potentiation [48,117,293,294,295].

## 8. Other Systems Contributing to the Pathogenesis of MDD

The cholinergic system is also implicated in the pathogenesis of depression because it was shown that the muscarinic cholinergic system is overactive or hyperresponsive in depression [77]. Numerous studies in humans [296,297] as well as animal models [298,299] indicate that hyperactive cholinergic system can be involved in the pathological process in depression [64,77,300]. Indeed, cholinergic receptors and neurons connect the septum with the hippocampus and VTA through interpeduncular nucleus and, thus, participate in the functioning of the reward system (see Figure 8 in a review by Loonen and Ivanova [149]). Some data also suggested that the muscarinic receptor system mediates the effects of cholinergic system on emotional behavior [77]. Some studies even specified that the M2 receptor might regulate mood in depression. Multiple polymorphisms in M2 receptor gene were linked with increased risk for developing MDD [301,302]. Acetylcholine is considered to play the central role in sensory and emotional processing; therefore, the overactive cholinergic system could change signal-to-noise processing, resulting in an overrepresentation of information laden with emotions and the creation of emotional processing bias correlated with cognitive deficiency in mood disorders [77].

Furthermore, nicotinic compounds may not only modulate mood and antidepressant action unidirectionally, but inhibition as well as activation of nicotinic acetylcholine receptors (nAChRs) may lead to antidepressant effects in different conditions. The nicotinic compounds affect different receptors, neurotransmitter systems, and brain areas, with diverse results in individuals experiencing different depressive symptoms or levels of stress [64]. Smoking is associated with depression, which indicates that smoking, namely nicotine intake, may affect the mood [303]. Low nicotine levels administered chronically (for example, by the nicotine patch) may desensitize nAChRs [304,305]; therefore, nAChRs blockade might be significant in the manifestation of the influence of nicotinic substances on depressive symptoms. Because acetylcholine is the endogenous neurotransmitter for nAChRs, and because nicotine affects depression, it can be concluded that the violation of regulation of the cholinergic system might be one of the triggers of MDD [64,306]. Though it has not been clarified how nicotinic substances can act as antidepressant-like agents yet, changes in the function of nAChR alone or in conjunction with monoamine-based antidepressants can become a new strategy in the treatment of mood disorders [64].

Decreased neurotransmission of monoamines could lead to the altered response of second messengers, even when levels of monoamine neurotransmitters are adequate [65]. Indeed, decreased levels of cyclic adenosine monophosphate and inositol were detected in the brains of patients with depression [307,308].

Histamine as a neurotransmitter also participates in the processes of arousal and wakefulness [309] and thus could play a role in the pathogenesis of depression, particularly, in changes of sleep/wake cycles. Histaminergic neurons mainly reside in the tuberomammillary nucleus (see Figure 3.24 in von Bohlen Und Halbach and Dermietzel [309]). The axons of histaminergic neurons reach various regions of the brain, such as the cerebellum, forebrain, mesencephalon, thalamic areas, nucleus accumbens, bed nuclei stria terminalis, the cerebral cortex, and hippocampus. However, no changes were found in the expression of histamine-related genes in depression [310].

## 9. One set of Intertwined Processes

In summary, many brain structures, neurotransmitters, hormones, and substances may be involved in the development of MDD. However, none of the hypotheses describing the development of depression are comprehensive because none of them can explain all the cases and mechanisms. The analysis of all hypotheses suggests that there is interdependence between the brain structures and various substances involved in the pathogenesis of MDD, which could be not entirely universal, but can affect all brain regions, to one degree or another (Figure 6), depending on the triggering factor, which, in turn, could explain the different subtypes of MDD. The analysis of reports and reviews presented above demonstrates that some common brain structures, such as amygdala (mainly BNST), hippocampus (neurogenesis and neurotrophins), PFC, and hypothalamus, and their interactions through neurotransmitters and biologically active substances (5-HT, NA, CRF) are characteristic for all theories of pathogenesis of MDD. The only difference between the theories could be triggering factors in each particular case and the subsequent cascade of events, which again would occur in those structures described above.

Over the past decades, it has become clear that the roles of stress and inflammation in the development of MDD are obvious. They can disrupt the chemical balance of a normal brain function (Figure 3, Figure 4 and Figure 6), as an influence of stress and inflammation in the form of proinflammatory cytokines and the subsequent chain of events and/or a destabilizing effect of stress on neurons in the PFC. Most individuals manage to recover to the normal state after the elimination of stress. However, if a damaging factor is persistent enough, the chemical balance of a brain shifts to a new self-sustaining state, which causes the manifestation of depressive symptoms [31]. For example, chronic stress leads to alterations in PFC and amygdala, shifting the balance of neurotransmitters in amygdala toward a depressive mood and inhibiting locus ceruleus and raphe nucleus. In addition, stress alters the levels of central cytokines, which, in turn, disrupt neurogenesis in hippocampus and initiate the pathological activation of the HPA axis. This axis may affect processes in suprachiasmatic nucleus and sleep/wake cycles. These processes are most likely supported and/or amplified with altered or impaired feedback loops from the affected areas (Figure 6). This new self-sustaining state of the brain in MDD includes the altered levels of NA and 5-HT in afferents from the LC and RN, respectively, and the changed levels of neurotransmitters in feedback afferents to the LC and RN [275]. Decreased 5-HT signaling from the RN leads to decreased 5-HT input to various brain areas such as nucleus accumbens, amygdala, hippocampus, and PFC. Research over the last 50 years has provided extensive evidence showing that abnormal monoamine neuronal function is an important underlying pathology in MDD [65]. Imaging studies indicated that MDD is associated with abnormal metabolism in limbic and paralimbic structures of the PFC [71]. This abnormal metabolism is normalized in the amygdala and PFC in patients showing a persistent antidepressant response [311].

Theoretically, HPA axis hyperactivity and inflammation in adult patients with depression (including responses to trauma in early childhood) might also be a part of the same pathological process. On the one hand, HPA axis hyperactivity is an indication of the ineffective action of glucocorticoid hormones, which could result in the activation of the immune system. On the other hand, inflammation could stimulate the activity of the HPA axis through a direct action of cytokines on the brain and by inducing glucocorticoid resistance [104,180,312,313,314].

In summary, highly complex interactions exist between physiological, functional, social, and psychological factors [315]. Moreover, the human brain clearly remains plastic, i.e., responsive to intrinsic and extrinsic stimulation, throughout life, which provides a good basis for the successful treatment of MDD.

## 10. Concluding Remarks

On the whole, it is necessary to emphasize that depression is a heterogeneous disorder that involves a wide range of subtypes (e.g., melancholic, atypical, and psychotic), with distinct characteristics in terms of symptomatology, neurobiology, and physiological and endocrine functioning [106]. It is evident from the literature reviewed above that the multiplicity of symptoms related to depression most likely is the result of aberrations in different aspects of normal neural functions that can range from the molecular level up to the neural circuit [117]. There may exist several subtypes of MDD with different etiopathogenesis [103]. The observation that classical antidepressant medications only work on a subset of patients indicates that patients with depression express aberration in different neural processes [117] or, rather, in various parts of the same complex mechanism consisting of an extensive network of interconnected pathways. It was proposed that CSF homovanilinic acid, hyper-/hypocortisolism, and CSF cytokines and plasma tryptophan are possible biomarkers for subtyping MDD and related conditions [103]. This subtyping may lead to the development of new strategies of treatment, such as DA agonists, antagonists of CRF/arginine-vasopressin receptors, and anti-inflammatory agents, and their tailor-made uses [103], as well as, antioxidants, which was demonstrated in a randomized controlled trial [316]. Characterizing patients with MDD with an underlying elevated inflammatory profile alone may ultimately help health-care professionals to develop a more effective personalized treatment plan for treatment-resistant individuals [39,254,288], which can be based on standardized treatment [317] with modifications based on “results from different phase clinical trials,” as reviewed by Cai and co-authors [318]. This is also true in the case of endocrine disturbances such as changes in glutamatergic and GABAergic signaling in the CNS [288].

A complete baseline assessment of depressive symptoms prior to treatment allows building a patient-specific profile, which, in turn, may help to develop a more efficient therapeutic plan. Clarification of the previous history of medication is necessary for differentiation between unresolved symptoms, current health conditions, and the side effects of prior treatment. Comprehension of the nature, mechanisms, and degree of functional impairment can aid physicians in the formulation of more efficient personalized pharmacotherapy and regimens of treatment for each patient’s unique constellation of symptoms [275].

Therefore, the future of treatment of depression might consist in the use of combined strategies in patients who are nonresponsive to traditional monotherapy [64].

## Figures and Tables

**Figure 1 cells-10-01283-f001:**
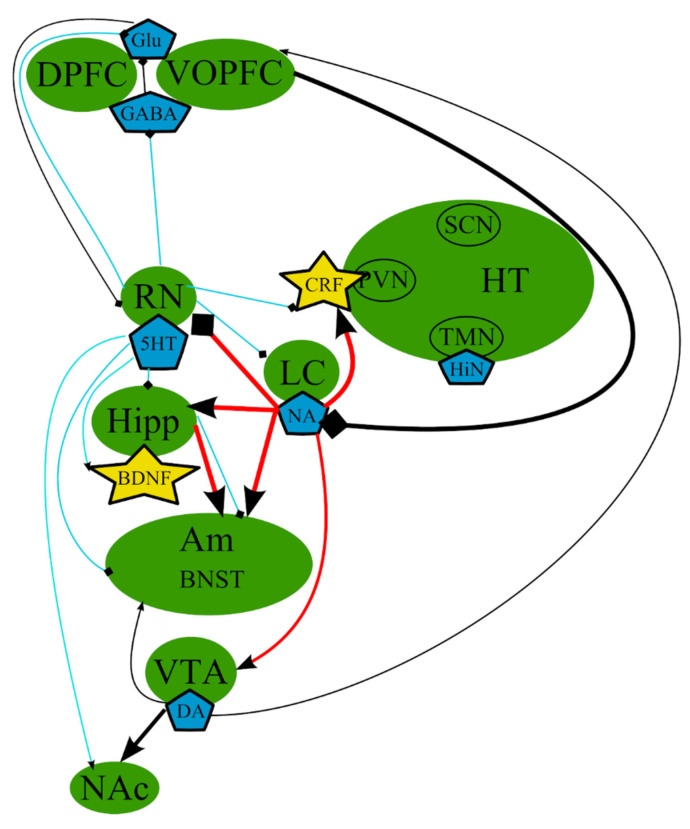
The monoamine hypothesis: main brain structures, neurotransmitters, biologically active substances, and interactions. BDNF—brain-derived neurotrophic factor; BNST—bed nuclei stria terminalis; CRF—corticotropin-releasing factor; DA—dopamine; DPFC—dorsal prefrontal cortex; Glu—glutamine; GABA—gamma-aminobutyric acid; HiN—histamine; Hipp—hippocampus; HT—hypothalamus; 5-HT—serotonin; LC—locus ceruleus; NA—noradrenaline; NAc—nucleus accumbens; Oxn—orexin; SCN—suprachiasmatic nucleus; TMN—tuberomammilar nucleus; RN—raphe nucleus; PVN—paraventricular nucleus; VOPFC—ventral and orbital prefrontal cortex; VTA—ventral tegmental area; -> (arrow): activating effect; -<> (rhombus): a black rhombus—inhibitory effect; thick line—effect is increased; thin line—effect is decreased; medium thickness line—effect is not changed, or alterations of the effect are unknown; red line—noradrenaline effect; blue line—serotonin effect; black line—various neurotransmitters or neuropeptides.

**Figure 2 cells-10-01283-f002:**
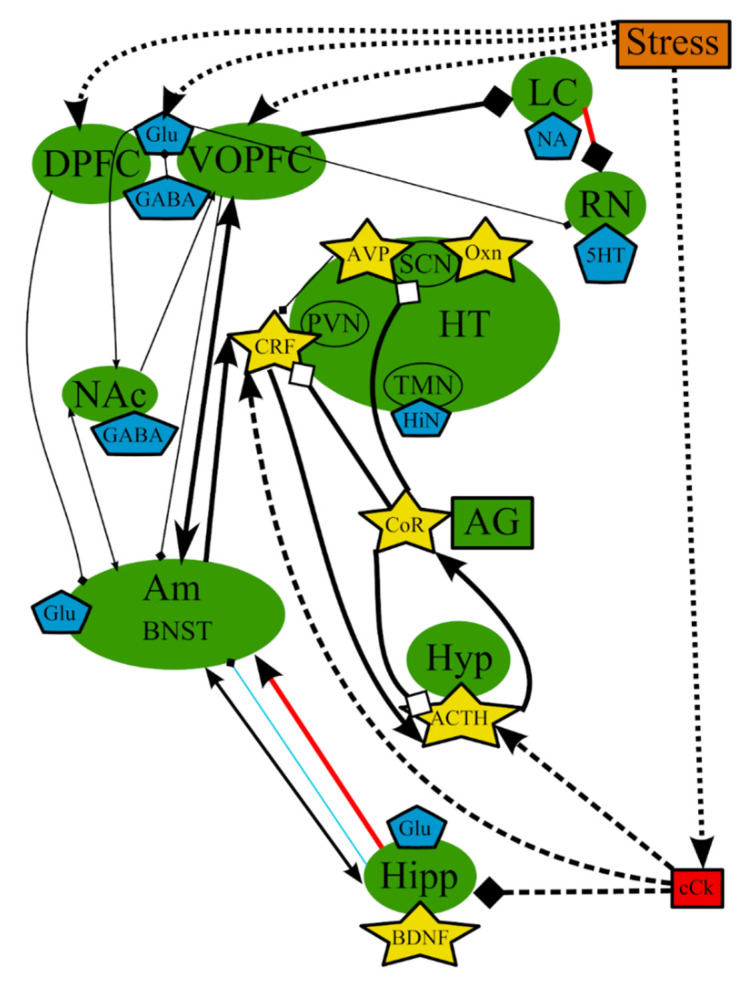
The hypothesis of stress-induced depression: main structures, neurotransmitters, biologically active substances, interactions, and external factors. ACTH—adrenocorticotropic hormone; AG—adrenal gland; Am—amygdala; AVP—arginine-vasopressin; BDNF—brain-derived neurotrophic factor; BNST—bed nuclei stria terminalis; cCk—central pro-inflammatory cytokines; CoR—cortisol; CRF—corticotropin-releasing factor; DA—dopamine; DPFC—dorsal prefrontal cortex; Glu—glutamine; GABA—gamma-aminobutyric acid; HiN—histamine; Hipp—hippocampus; HT—hypothalamus; 5-HT—serotonin; Hyp—hypophysis; LC—locus ceruleus; NA—noradrenaline; NAc—nucleus accumbens; Oxn—orexin; SCN—suprachiasmatic nucleus; TMN—tuberomammilar nucleus; RN—raphe nucleus; PVN—paraventricular nucleus; VOPFC—ventral and orbital prefrontal cortex; -> (arrow): activating effect; -<> (rhombus): a black rhombus—inhibitory effect; a white rhombus: an effect is blocked or ineffective because the receptor is not sensitive; thick line—effect is increased; thin line—effect is decreased; medium thickness line—effect is not changed or alterations of the effect are unknown; red line—noradrenaline effect (most of them were omitted to simplify the figure); blue line—serotonin effect (most of them were omitted to simplify the figure); black line—various neurotransmitters or neuropeptides; dotted lines—influence of external factors.

**Figure 3 cells-10-01283-f003:**
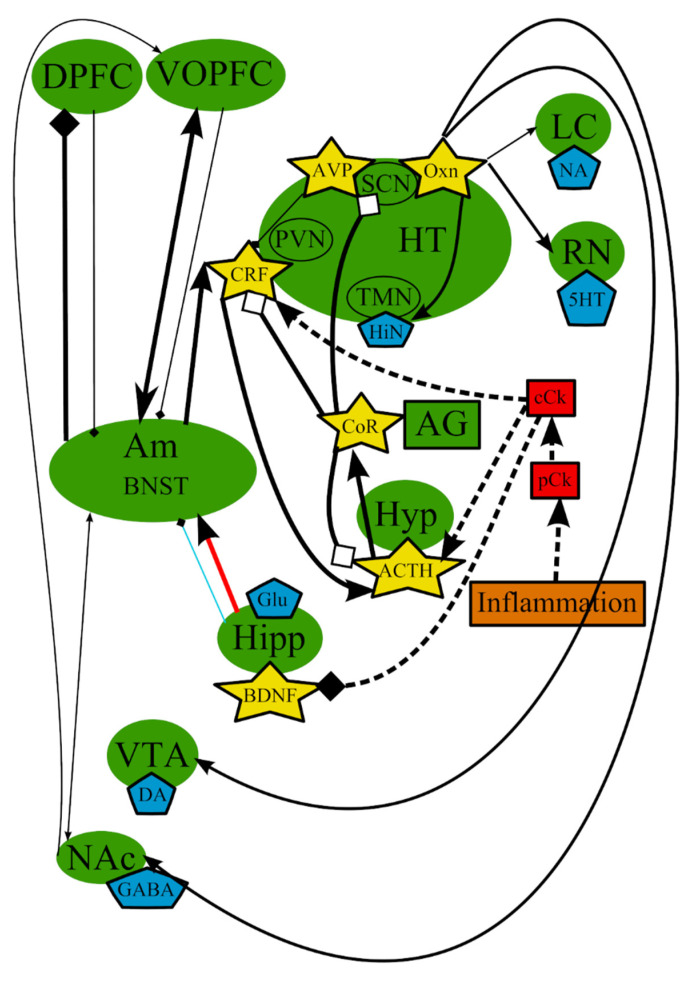
The inflammation/cytokine hypothesis: main structures, neurotransmitters, biologically active substances, interactions, and external factors. ACTH—adrenocorticotropic hormone; AG—adrenal gland; Am—amygdala; AVP—arginine-vasopressin; BDNF—brain-derived neurotrophic factor; BNST—bed nuclei stria terminalis; cCk—central pro-inflammatory cytokines; CoR—cortisol; CRF—corticotropin-releasing factor; DA—dopamine; DPFC—dorsal prefrontal cortex; Glu—glutamine; GABA—gamma-aminobutyric acid; HiN—histamine; Hipp—hippocampus; HT—hypothalamus; 5-HT—serotonin; Hyp—hypophysis; LC—locus ceruleus; NA—noradrenaline; NAc—nucleus accumbens; Oxn—orexin; SCN—suprachiasmatic nucleus; TMN—tuberomammilar nucleus; RN—raphe nucleus; pCk—peripheral pro-inflammatory cytokines; PVN—paraventricular nucleus; VOPFC—ventral and orbital prefrontal cortex; VTA—ventral tegmental area; -> (arrow): activating effect; -<> (rhombus): a black rhombus—inhibitory effect; a white rhombus: an effect is blocked or ineffective because the receptor is not sensitive; thick line—effect is increased; thin line—effect is decreased; medium thickness line—effect is not changed or alterations of the effect are unknown; red line—noradrenaline effect (most of them were omitted to simplify the figure); blue line—serotonin effect (most of them were omitted to simplify the figure); black line—various neurotransmitters or neuropeptides; dotted lines—influence of external factors.

**Figure 4 cells-10-01283-f004:**
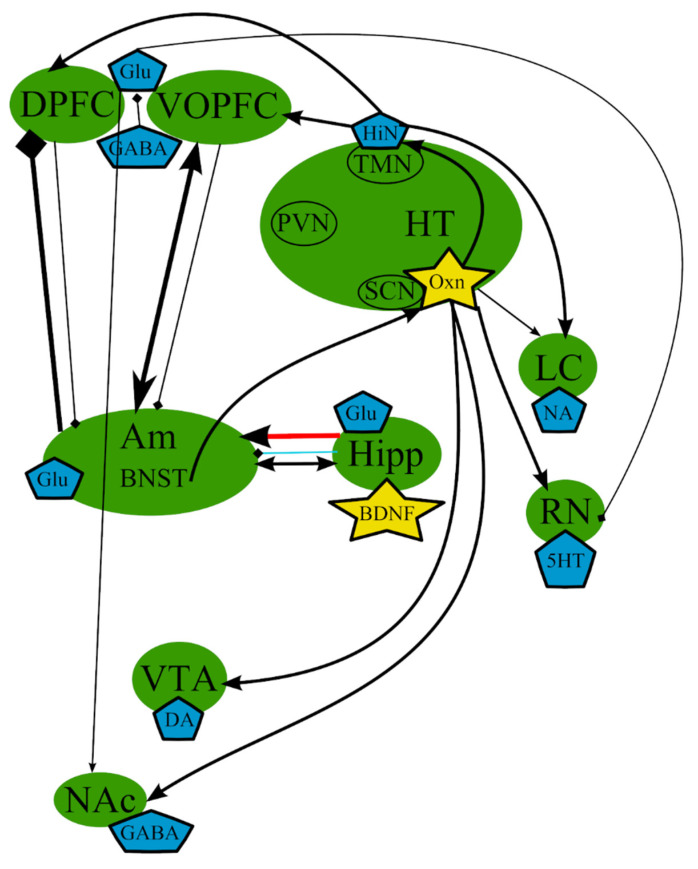
The circadian hypothesis: main brain structures, neurotransmitters, biologically active substances, and interactions. Am—amygdala; AVP—arginine-vasopressin; BDNF—brain-derived neurotrophic factor; BNST—bed nuclei stria terminalis; CRF—corticotropin-releasing factor; DA—dopamine; DPFC—dorsal prefrontal cortex; Glu—glutamine; GABA—gamma-aminobutyric acid; HiN—histamine; Hipp—hippocampus; HT—serotonin; Hyp—hypophysis; LC—locus ceruleus; NA—noradrenaline; NAc—nucleus accumbens; Oxn—orexin; SCN—suprachiasmatic nucleus; TMN—tuberomammilar nucleus; RN—raphe nucleus; PVN—paraventricular nucleus; VOPFC—ventral and orbital prefrontal cortex; VTA—ventral tegmental area; -> (arrow): activating effect; -<> (rhombus): a black rhombus—inhibitory effect; thick line—effect is increased; thin line—effect is decreased; medium thickness line—effect is not changed or alterations of the effect are unknown; red line—noradrenaline effect (most of them were omitted to simplify the figure); blue line—serotonin effect (most of them were omitted to simplify the figure); black line—various neurotransmitters or neuropeptides.

**Figure 5 cells-10-01283-f005:**
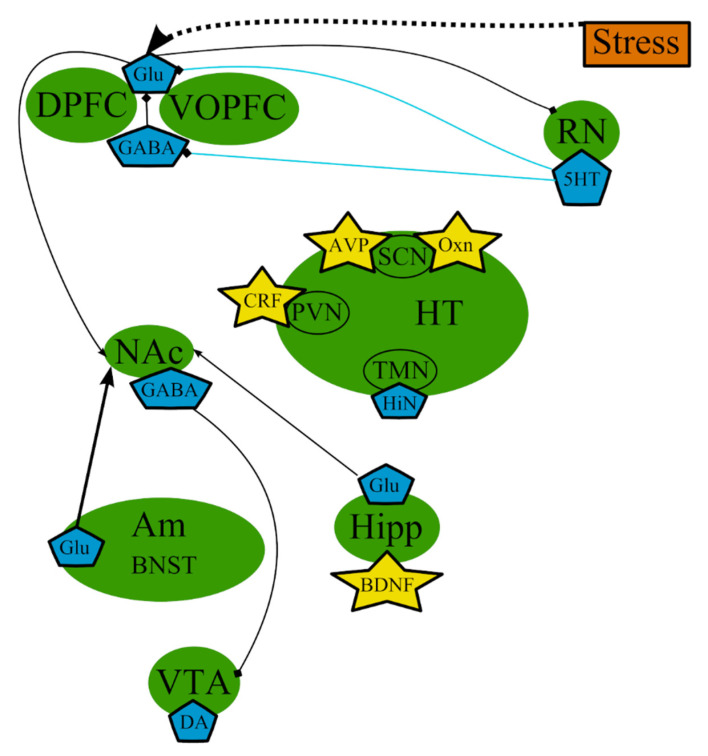
The excitatory neurotransmitters involved in the pathogenesis of MDD: main brain structures, neurotransmitters, biologically active substances, interactions, and external factors. Am—amygdala; AVP—arginine-vasopressin; BDNF—brain-derived neurotrophic factor; BNST—bed nuclei stria terminalis; CRF—corticotropin-releasing factor; DA—dopamine; DPFC—dorsal prefrontal cortex; Glu—glutamine; GABA—gamma-aminobutyric acid; HiN—histamine; Hipp—hippocampus; HT—hypothalamus; 5-HT—serotonin; NA—noradrenaline; NAc—nucleus accumbens; Oxn—orexin; SCN—suprachiasmatic nucleus; TMN—tuberomammilar nucleus; RN—raphe nucleus; PVN—paraventricular nucleus; VOPFC—ventral and orbital prefrontal cortex; VTA—ventral tegmental area; -> (arrow): activating effect; -<> (rhombus): a black rhombus—inhibitory effect; thick line—the effect is increased; thin line—the effect is decreased; medium thickness line—the effect is not changed or alterations of the effect are unknown; blue line—serotonin effect (most of them were omitted to simplify the figure); black line—various neurotransmitters or neuropeptides.

**Figure 6 cells-10-01283-f006:**
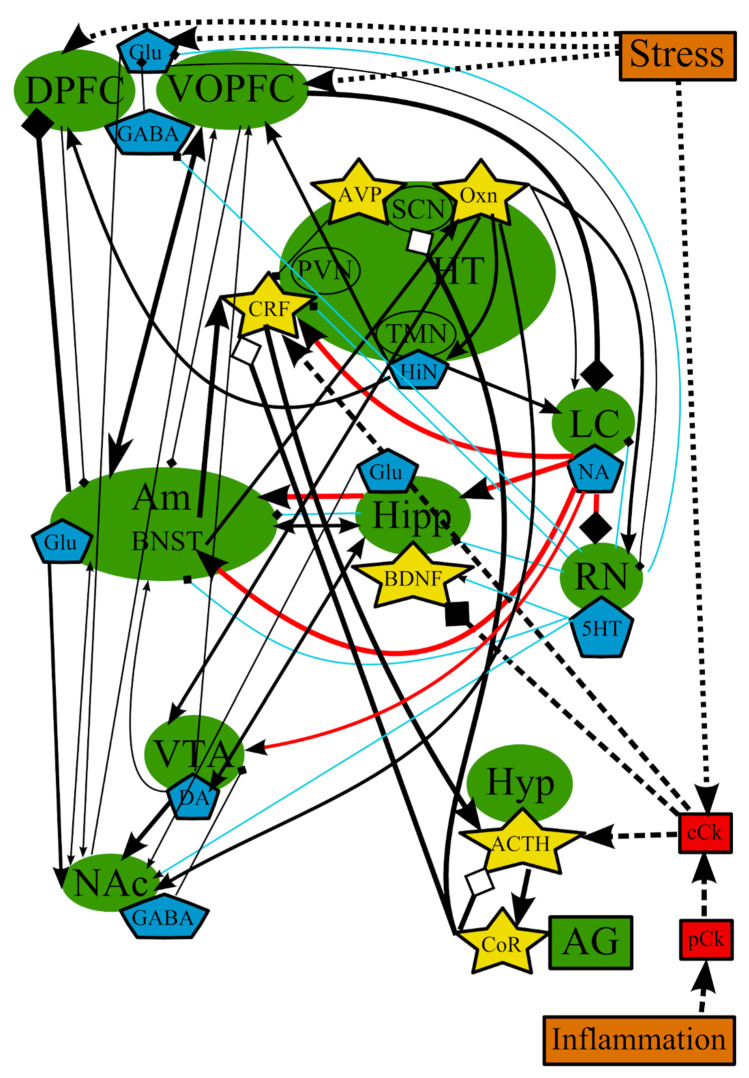
The overall picture of the combined common mechanisms and relationships of all major theories of depression. ACTH—adrenocorticotropic hormone; AG—adrenal gland; Am—amygdala; AVP—arginine-vasopressin; BDNF—brain-derived neurotrophic factor; BNST—bed nuclei stria terminalis; cCk—central pro-inflammatory cytokines; CoR—cortisol; CRF—corticotropin-releasing factor; DA—dopamine; DPFC—dorsal prefrontal cortex; Glu—glutamine; GABA—gamma-aminobutyric acid; HiN—histamine; Hipp—hippocampus; HT—hypothalamus; 5-HT—serotonin; Hyp—hypophysis; LC—locus ceruleus; NA—noradrenaline; NAc—nucleus accumbens; Oxn—orexin; SCN—suprachiasmatic nucleus; TMN—tuberomammilar nucleus; RN—raphe nucleus; pCk—peripheral pro-inflammatory cytokines; PVN—paraventricular nucleus; VOPFC—ventral and orbital prefrontal cortex; VTA—ventral tegmental area; -> (arrow): activating effect; -<> (rhombus): a black rhombus—inhibitory effect; a white rhombus—an effect is blocked or ineffective because the receptor is not sensitive; thick line—effect is increased; thin line—effect is decreased; medium thickness line—the effect is not changed or alterations of the effect are unknown; red line—noradrenaline effect; blue line—serotonin effect; black line—various neurotransmitters or neuropeptides; dotted lines—influence of external factors.

**Table 1 cells-10-01283-t001:** The major theories of depression.

Theory	Probable Cause	Structures Involved	Neuro-Transmitters and BAS, Which Levels Are Altered	Weaknesses of Theory
The monoamine hypothesis [65]	Genetic vulnerability; stress; environmental vulnerability	**Am (BNST** ^a^); **DPFC, VOPFC**^a^; LC; RN; **Hipp**	**NA**^a^; **5-HT**^a^; **CRF**^a^	Not all causes and symptoms are explained
The hypothesis of stress-induced depression [66]	Stress; genetic susceptibility	HT (SCN ^a^; PVN ^a^); pituitary gland; adrenal glands; **Am (BNST)**; **DPFC, VOPFC**; **Hipp**	**CRF**^a^; **cortisol**; **AVP**^a^; ACTH ^a^; Oxn ^a^; BDNF ^a^	Does not explain all cases; there is no single mechanism that explains all alterations in HPA axis
The inflammation/cytokine hypothesis [39]	Stress; inflammation; genetic susceptibility	HT (SCN; PVN); pituitary gland; adrenal glands; **Am (BNST)**; **DPFC, VOPFC**; **Hipp** (microglia activation); LC; RN	pCk ^a^; cCk ^a^; **CRF**; **cortisol**; **AVP**; ACTH; **5-HT**; **NA**; BDNF	Does not explain all cases
The neurotrophic hypothesis [67]	Stress; inflammation; genetic susceptibility	**Hipp**; **PFC**; HT (SCN; PVN); pituitary gland; adrenal glands; **Am (BNST)**; **LC**	cCk; Glu ^a^; GABA ^a^; **CRF**; **cortisol**; **AVP**; ACTH; **NA**; **5-HT**; BDNF	Does not explain all cases; does not provide adequate mechanism of the development of the disease
The GABA-glutamate-mediated hypothesis [56]	Genetic susceptibility; environmental vulnerability; possibly stress	**DPFC, VOPFC**; LC; RN; **Am (BNST)**; **Hipp**	Glu; GABA; **5-HT**; **NA**	Does not explain causes of the disease; does not provide adequate mechanism of the development of the disease
The circadian hypothesis [68]	Stress; continuously altered/irregular diurnal cycle; possibly evening chronotype; genetic susceptibility	HT (SCN; PVN); pituitary gland; adrenal glands; **Am (BNST)**; **DPFC**, **VOPFC**; **Hipp**; LC; RN	Oxn; possibly melatonin; **CRF**; **cortisol**; **AVP**; ACTH; **5-HT**; **NA**	Does not explain all cases; the primary cause is unclear

^a^ AVP—arginine-vasopressin; ACTH—adrenocorticotropin hormone; Am—amygdala; BAS—biologically active substances; BDNF—brain-derived neurotrophic factor; BNST—bed nuclei stria terminalis; DPFC—dorsal prefrontal cortex; GABA—gamma-aminobutyric acid; Glu—glutamate; Hipp—hippocampus; HT—hypothalamus; NA—noradrenalin; 5-HT—serotonin; CRF—corticoliberin; LC—locus ceruleus; RN—raphe nucleus; SCN—suprachiasmatic nucleus; PVN—paraventricular nucleus; Oxn—orexin; pCk—peripheral pro-inflammatory cytokines; cCk—central pro-inflammatory cytokines; VOPFC—ventral and orbital prefrontal cortex. The common structures and neuro-transmitters and BAS are given in bold.

## Data Availability

No new data were created or analyzed in this study. Data sharing is not applicable to this article.

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
