# Peer review of "Major Depression: One Brain, One Disease, One Set of Intertwined Processes"

_cells, 2021, doi:10.3390/cells10061283_

Round 1

Reviewer 1 Report

The review manuscript by Filatova et al. entitled “Major depression: one brain, one disease, one set of intertwined processes” is interesting. However, there are following points:

  1. Give the details whether the study was clinical or pre-clinical (Throughout the manuscript). Some places authors mentioned this but most of the place’s reader has to figure it out. For example, Lines 204-206, “Chronic stress…….basolateral amygdala [128,129]”. In this statement its not clear whether human or animal study. Overall, for pre-clinical studies add the details of models used. For clinical study give the details of patient background and cohort etc.
  2. Lines #152-154. “Overall, numerous studies suggest……depressive disorders[105].” Authors say numerous studies and provided only one refence. Please rephrase or add more refences. Again Lines 399-400 “Some studies demonstrated………with MDD [277].” This statement also states some studies and only one reference. Please rephrase or add more refences. Do it throughout the manuscript.
  3. Line #90. Its written Table-2, it should be table-1
  4. Give a title to the table
  5. Add refences in the table
  6. Please add legend for table
  7. Lot of places there are no connection between the paragraphs. Try to connect each paragraph and topics for the readers ease.
  8. Review manuscript lack recent refences/updates. There are no references from the year 2019, 2020 and 2021.
  9. Give some update on clinical trials with these treatments. Their success rate or failure or current status.
  10. Conclusion is too long. Make it short and to the point.

Author Response

We would like to thank the reviewer for the thorough review of our manuscript. Below there are the comments of the reviewers in the font “Times New Roman” and our responses to their comments in the font “Courier new”.

The review manuscript by Filatova et al. entitled “Major depression: one brain, one disease, one set of intertwined processes” is interesting. However, there are following points:

Give the details whether the study was clinical or pre-clinical (Throughout the manuscript). Some places authors mentioned this but most of the place’s reader has to figure it out. For example, Lines 204-206, “Chronic stress…….basolateral amygdala [128,129]”. In this statement it’s not clear whether human or animal study. Overall, for pre-clinical studies add the details of models used. For clinical study give the details of patient background and cohort etc.

We corrected this issue as best as we could, however, in most cases we refer to other reviews, which cover various studies themselves, and it is almost impossible to give such details without overcomplication of the manuscript. Below are some examples of our corrections:

“…which results in adrenal hypertrophy and thymic atrophy associated with long exposure to corticotropin and elevated  glucocorticoid hormone in rats[85]” later in “3. The hypothesis of stress-induced depression” section.

Or “…that form synaptic connections in mice approximately in 4 weeks[116,138]…” and “…and medial amygdala and increased spine density in the basolateral amygdala in animal studies[130, 131]” in “4. The neurotrophic hypothesis” section.

“Several clinical and animal studies reported that proinflammatory cytokines could be involved in the development of MDD [39,40,155-159]” in “5. The inflammation/cytokine hypothesis” section.

Lines #152-154. “Overall, numerous studies suggest……depressive disorders[105].” Authors say numerous studies and provided only one reference. Please rephrase or add more references. Again Lines 399-400 “Some studies demonstrated………with MDD [277].” This statement also states some studies and only one reference. Please rephrase or add more references.

We corrected this issue. The following sentences were rephrased as follows:

“Overall, numerous studies, reviewed by Willkinson and Goodyer, suggest that continuous dysregulation of the HPA axis with a central deficit of the feedback mechanisms is predominant in depressive disorders[106].”

“Despite numerous evidence demonstrating stress-induced decrease of neurogenesis, several studies, reviewed by Hanson et ali, have also shown the lack of correlation between stress and neurogenesis [144].”

“Some studies, reviewed by Hasler et ali, demonstrated abnormally decreased plasma and CSF levels of GABA in patients with MDD [284].”

Line #90. Its written Table-2, it should be table-1

We corrected this issue and renamed the table as follows: “Table 1.”

Give a title to the table

We corrected this issue and renamed the table as follows: “Table 1. The major theories of depression.”

Add references in the table

We corrected this issue and added some references in the table.

Please add legend for table

The legend for the table was present as a footnote in the original manuscript. However, we added a reference to this footnote in the table after the first mention of a term in the table.

Lot of places there are no connection between the paragraphs. Try to connect each paragraph and topics for the readers ease.

We corrected this issue. For example, the first sentence of “3. The hypothesis of stress-induced depression” section was modified as follows: “The hypothesis of stress-induced depression was the first hypothesis that aimed to explain possible causes of MDD, which were not clarified by the monoamine hypothesis”;

Or

“In addition to previously described processes other mechanisms of the MDD etiopathogenesis were proposed. They include inflammation[39]and microglial cells [1,151-154]” in “5. The inflammation/cytokine hypothesis” section.

“In addition, several primary studies and comprehensive reviews…” at the end of “5. The inflammation/cytokine hypothesis” section.

Review manuscript lack recent references/updates. There are no references from the year 2019, 2020 and 2021.

We updated the list of references where it was appropriate. We included references 14, 17, 121, 149, 150, 159, 245, 253, 286, 288, 317, 318.

Give some update on clinical trials with these treatments. Their success rate or failure or current status.

We corrected the manuscript by providing a reference to the latest review of recent clinical trials of new promising drugs. The correction can be found in the “Concluding remarks” section: “… which can be based on standardized treatment [317] with modifications based on “results from different phase clinical trials” reviewed by Cai and co-authors[318]”.

Conclusion is too long. Make it short and to the point.

We corrected this issue by introducing additional section “9. One set of intertwined processes”, where we summarize all the theories described, and reducing the Conclusion.

Reviewer 2 Report

The review article entitled “Major depression: one brain, one disease, one set of intertwined processes “ by Filatova and colleagues aims to review the current state of knowledge regarding the neurobiological mechanisms involved in the development of major depressive disorders. For this, the authors briefly described the different pathophysiological mechanisms (i.e. alterations in monoamines, HPA axis function, inflammatory processes, changes in neurotrophin levels, imbalance in excitatory neurotransmitters and abnormal circadian rhythms) involved in major depressive disorders. As quite nicely reviewed, major depressive disorders are thought to result from the complex interplay of multiple inherited genetic factors and subsequent exposure to environmental stressors throughout life. The exact involvement of each pathophysiological mechanism described continue to be defined.

In this review article, the authors aimed to present the strengths and weaknesses of each central pathophysiological mechanism involved in major depressive disorders. In addition, they aimed to make parallels between the different pathophysiological mechanisms, in order to highlight common alterations and combine them into a framework.

To my opinion, the topic of the review is interesting and summarizes well our current understanding of the different pathophysiological mechanisms involved in major depressive disorders. It is also very well-written, focused and comprehensively described.

However, the review in its current state lacks novelty and originality. It is mainly due a highly descriptive (rather than integrative) content (brief summary of each theory, presentations of some strengths and weaknesses).

The review lacks a clearly defined structure for each theory (for example, the authors could describe the bases of the theory, recapitulate the data in humans, compare with the data in rodents, present the main strengths and weaknesses, and write a short conclusion). This will help to be much more consistent for each major theory.

Moreover, the review needs to provide a critical discussion that explicitly explains the common mechanisms and relationships between the different pathophysiological mechanisms described (i.e. how to reconcile the alterations in monoamines, HPA axis function, inflammatory processes, changes in neurotrophin levels, imbalance in excitatory neurotransmitters and abnormal circadian rhythms involved in major depressive disorders?) –the initial aim of the review. Here, no clear picture emerges on the common mechanisms and relationships of all major theories of major depressive disorders.

Furthermore, the table in its current form has not much interest (it could be revised to nicely highlight the common causes, brain regions and neurotransmitters!); the figure is barely understandable as very complex and messy. Although such figure is a good idea, it should be extensively revised to give a clear picture of the common mechanisms and relationships of all major theories of major depressive disorders.

I would then suggest the author to provide some major revisions to their review.

Author Response

We would like to thank the reviewer for the thorough review of our manuscript. Below there are the comments of the reviewers in the font “Times New Roman” and our responses to their comments in the font “Courier new”.

Reviewer 2.

The review article entitled “Major depression: one brain, one disease, one set of intertwined processes “ by Filatova and colleagues aims to review the current state of knowledge regarding the neurobiological mechanisms involved in the development of major depressive disorders. For this, the authors briefly described the different pathophysiological mechanisms (i.e. alterations in monoamines, HPA axis function, inflammatory processes, changes in neurotrophin levels, imbalance in excitatory neurotransmitters and abnormal circadian rhythms) involved in major depressive disorders. As quite nicely reviewed, major depressive disorders are thought to result from the complex interplay of multiple inherited genetic factors and subsequent exposure to environmental stressors throughout life. The exact involvement of each pathophysiological mechanism described continue to be defined.

In this review article, the authors aimed to present the strengths and weaknesses of each central pathophysiological mechanism involved in major depressive disorders. In addition, they aimed to make parallels between the different pathophysiological mechanisms, in order to highlight common alterations and combine them into a framework.

To my opinion, the topic of the review is interesting and summarizes well our current understanding of the different pathophysiological mechanisms involved in major depressive disorders. It is also very well-written, focused and comprehensively described.

However, the review in its current state lacks novelty and originality. It is mainly due a highly descriptive (rather than integrative) content (brief summary of each theory, presentations of some strengths and weaknesses).

To our knowledge there has not been published any review which covers all (not some) theories and hypothesis of development of MDD. In this regard our manuscript and the main figure (Figure 6) are unique.  

The review lacks a clearly defined structure for each theory (for example, the authors could describe the bases of the theory, recapitulate the data in humans, compare with the data in rodents, present the main strengths and weaknesses, and write a short conclusion). This will help to be much more consistent for each major theory.

We added several figures to better describe structures of theories. Our manuscript initially contained Table 1 to address the issue of main strengths and weaknesses. We also briefly describe each theory in the relevant section and refer to excellent and more specific reviews, which describe each theory in much greater detail. Comparison of studies on humans and animals are presented in the reviews we refer to, so we believe it would be redundant to include such data in our manuscript.

Moreover, the review needs to provide a critical discussion that explicitly explains the common mechanisms and relationships between the different pathophysiological mechanisms described (i.e. how to reconcile the alterations in monoamines, HPA axis function, inflammatory processes, changes in neurotrophin levels, imbalance in excitatory neurotransmitters and abnormal circadian rhythms involved in major depressive disorders?) – the initial aim of the review. Here, no clear picture emerges on the common mechanisms and relationships of all major theories of major depressive disorders.

We have introduced Figures 1-5 to illustrate separate theories and to demonstrate how Figure 6 emerges from combination of these figures. We also introduced additional section “9. One set of intertwined processes”, where we give an example of combination of the theories described.

Furthermore, the table in its current form has not much interest (it could be revised to nicely highlight the common causes, brain regions and neurotransmitters!); the figure is barely understandable as very complex and messy. Although such figure is a good idea, it should be extensively revised to give a clear picture of the common mechanisms and relationships of all major theories of major depressive disorders.

We revised Table 1: highlighted common causes, structures, and biologically active substances, deleted some unnecessary text, and added some references.

The Figure 6 already reflects all common mechanisms. Human brain is a very complex department of the nervous system, and MDD affects many of its structures. All of these common mechanisms are intertwined and it seems impossible to separate one from another, especially when taking into account the fact that one process may influence several others which, in turn, alone or in combination, could affect the first one in a feedback-loop manner. However, we attempted to simplify the Figure 6 by providing simplified figures for most of the theories, which were made on the basis of Figure 6. We believe that it should be easier to trace separate processes from Figures 1-5 on Figure 6.

I would then suggest the author to provide some major revisions to their review.

Round 2

Reviewer 1 Report

Authors' have satisfactorily revised the manuscript. 

Author Response

We would again like to thank the reviewer for the thorough review of our manuscript. 

Reviewer 2 Report

The manuscript provides a good review of the different theories and hypotheses of major depression but failed to achieve its initial goal, i.e. providing a clear picture of the common mechanisms and relationships of all theories of major depression. Althought the table and the figures have been simplified and improved, there is still quite a  lot of work to do on the figures. Please avoid lines crossing and non necessary lines (no effects or unchanged), and provide a brief but clear description of the figure in the legend (as said, almost no explanation is provided clearly for the molecular mechanisms drawn in each figure!). A critical discussion of common mechanisms and relationships of all theories of major depression is still crucially missing, to my opinion, to make this review verw original and attractive to the readers. I therefore recommend again major revisions.

Author Response

We would again like to thank the reviewer for the thorough review of our manuscript. Below there are the comments of the reviewers in the font “Times New Roman” and our responses to their comments in the font “Courier new”.

We modified the figures as best we could. However, it was impossible to avoid lines crossing on some of them. However, we believe that the lines showing the unchanged or unknown effect are necessary because they demonstrate an important link between structures. Therefore, we did not remove them. We also added brief description of the figures in the legends.

We also added some discussion of common mechanisms and relationships of all theories of MDD to the first paragraph of the “9. One set of intertwined processes” section.

Round 3

Reviewer 2 Report

I appreciate that my comments have been integrated for most of them. I think that the review is more readable now and will be an interesting read for those interested in the neurobiological bases of major depression. Good work!